# Antibodies against medically relevant arthropod-borne viruses in the ubiquitous African rodent *Mastomys natalensis*

**Wim De Kesel**[1,2]*, **Bram Vanden Broecke**[1,3], **Benny Borremans**[1,4], **Léa Fourchault**[5], **Elisabeth Willems**[2], **Ann Ceulemans**[2,6], **Christopher Sabuni**[7], **Apia Massawe**[7], **Rhodes H. Makundi**[7], **Herwig Leirs**[1], **Martine Peeters**[8], **Erik Verheyen**[1,5], **Sophie Gryseels**[1,5], **Joachim Mariën**[1,6], **Kevin K. Ariën**[2,9]*

1 Evolutionary Ecology Group, Department of Biology, Faculty of Science, University of Antwerp, Antwerp, Belgium, 2 Virology Unit, Department of Biomedical Sciences, Institute of Tropical Medicine, Antwerp, Belgium, 3 Terrestrial Ecology Unit, Department of Biology, Ghent University, Ghent, Belgium, 4 Wildlife Health Ecology Research Organization, San Diego, California, United States of America, 5 OD Taxonomy & Phylogeny, Royal Belgian Institute of Natural Sciences, Brussels, Belgium, 6 Virus Ecology Unit, Department of Biomedical Sciences, Institute of Tropical Medicine, Antwerp, Belgium, 7 Institute of Pest Management, Sokoine University of Agriculture, Morogoro, Tanzania, 8 TransVIHMI, University of Montpellier, Institut de Recherche pour le Développement (IRD), INSERM, Montpellier, France, 9 Department of Biomedical sciences, Faculty of Pharmaceutical, Biomedical and Veterinary Sciences, University of Antwerp, Antwerp, Belgium

* wim.dekesel@uantwerpen.be (WDK); karien@itg.be (KKA)

**Data Availability Statement:** All relevant data are within the manuscript and its Supporting Information files.

## Abstract

Over the past decades, the number of arthropod-borne virus (arbovirus) outbreaks has increased worldwide. Knowledge regarding the sylvatic cycle (i.e., non-human hosts/environment) of arboviruses is limited, particularly in Africa, and the main hosts for virus maintenance are unknown. Previous studies have shown the presence of antibodies against certain arboviruses (i.e., chikungunya-, dengue-, and Zika virus) in African non-human primates and bats. We hypothesize that small mammals, specifically rodents, may function as amplifying hosts in anthropogenic environments. The detection of RNA of most arboviruses is complicated by the viruses' short viremic period within their hosts. An alternative to determine arbovirus hosts is by detecting antibodies, which can persist several months. Therefore, we developed a high-throughput multiplex immunoassay to detect antibodies against 15 medically relevant arboviruses. We used this assay to assess approximately 1,300 blood samples of the multimammate mouse, *Mastomys natalensis* from Tanzania. In 24% of the samples, we detected antibodies against at least one of the tested arboviruses, with high seroprevalences of antibodies reacting against dengue virus serotype one (7.6%) and two (8.4%), and chikungunya virus (6%). Seroprevalence was higher in females and increased with age, which could be explained by inherent immunity and behavioral differences between sexes, and the increased chance of exposure to an arbovirus with age. We evaluated whether antibodies against multiple arboviruses co-occur more often than randomly and found that this may be true for some members of the *Flaviviridae* and *Togaviridae*. In conclusion, the development of an assay against a wide diversity of medically relevant arboviruses enabled the analysis of a large sample collection of one of the most abundant

**Funding:** This study was funded by The Research Foundation – Flanders (FWO) through the Senior research project G054820N (to KKA, EV and MP) and PhD fellowship 1171023N (to WDK) (https://www.fwo.be/en/). WDK received a salary from the Senior research project G054820N for the year 2020-2021 and is currently paid from the PhD fellowship 1171023N. The funders had no role in study design, data collection and analysis, decision to publish, or preparation of the manuscript.

**Competing interests:** The authors have declared that no competing interests exist.

African small mammals. Our findings highlight that *Mastomys natalensis* is involved in the transmission cycle of multiple arboviruses and provide a solid foundation to better understand the role of this ubiquitous rodent in arbovirus outbreaks.

## Author summary

One of the main causes of zoonotic related human morbidity and mortality is the transmission of arthropod-borne viruses such as dengue virus, Yellow Fever virus, and chikungunya virus. These viruses cannot only infect humans but also livestock, pets, and wildlife, though our understanding of their non-human hosts remains limited. Rodents are thought to be an important host for these viruses because they can be abundant, often live near humans, and some are already known to be viral hosts. However, research has focused mostly on non-human primates, neglecting other potential wild hosts. To address this gap, we have developed a high-throughput antibody test to screen rodent blood against 15 different arboviruses. Our findings reveal that *Mastomys natalensis*, a common African rodent species, carries antibodies that (cross-)react against these viruses. We hypothesize that immunologically naïve juveniles may drive transmission, particularly during population outbreaks. These outbreaks coincide with environmental conditions that are favorable for mosquitoes, thus increasing the risk of spillover to humans, livestock, and wildlife. Understanding the role of rodents in arbovirus transmission dynamics is crucial for mitigating zoonotic disease risks.

## Introduction

The African continent harbors a diverse array of infectious diseases with profound impacts on public health, economic development, and general well-being [1,2]. Diseases caused by arthropod-borne viruses, collectively known as arboviruses, are a growing threat for Africa and the rest of the world especially in relation to climate and environmental changes [3,4]. Arboviruses are a polyphyletic clade that includes several viral families, of which the most important are *Flaviviridae*, *Togaviridae*, *Bunyaviridae*, and *Reoviridae* [5]. Some well-known arboviruses, notorious for their negative effects on human health, are dengue virus, Yellow Fever virus, Zika virus, and chikungunya virus. Mosquitoes, ticks, sandflies, and midges are the primary vectors responsible for arbovirus transmission as they engage in hematophagy. These vectors do not only affect humans and livestock, but also a wide range of wildlife hosts [6–8]. Indeed, while for some arboviruses morbidity and mortality can be high in humans, similar impacts have been detected in other animals by arboviruses such as Rift Valley Fever virus in goats and sheep, West Nile virus in birds and horses, and Japanese Encephalitis virus in birds and pigs [4,5,9–11]. The (re-)emergence of arboviruses is linked to increased urbanization and global connectivity, natural genetic evolution of viruses, and adaptations of the vectors to changing climate and environments [11,12]. Emerging arboviruses pose a threat for humans, livestock, as well as wildlife, therefore it needs to be approached from a One health perspective (i.e., including human, animal, and environmental health) [13,14]. Nevertheless, our knowledge about the extent to which wild animals can serve as sylvatic hosts for human-infecting arboviruses and the natural diversity of arboviruses remains insufficient. This significantly limits our understanding of arbovirus transmission dynamics, which is required to develop more effective control measurements.

For decades, efforts have been made to identify natural reservoirs of arboviruses to monitor, prevent, and control sources of infection that pose a threat to human health [15–17]. Several studies have proposed non-human primates as significant potential reservoirs for arboviruses, as they have found arbovirus antibodies and viral RNA in this animal group [18–20]. However, other animal groups such as small mammals have often been neglected as potential arbovirus hosts [21]. Sporadic reports of arboviruses in small mammal species suggest that a more comprehensive investigation of their potential role as a host is needed [22–24].

Rodents have a number of characteristics that could make them an important hosts for several pathogens, including arboviruses [23]. Particularly the high species diversity, the fact that many species can reach high population abundances, and turnover rates. The risk of pathogen spillover to humans increases with the role of some rodents as a pest species, due to their proximity to humans [25,26]. A notable example of such a pest species is the ubiquitous rodent *Mastomys natalensis*, commonly known as the multimammate mouse. This species inhabits many regions of sub-Saharan Africa, with a preference for crop fields, fallow land, and typically occurring within or at the fringes of urban settlements [27,28]. In east Africa, especially in Tanzania, the reproductive cycle of *M. natalensis* is strongly correlated with seasonal rainfall which leads to strong seasonal fluctuations in density (20–500 individuals/hectare) and occasionally even severe population outbreaks (>1000 individuals/hectare) [29–32]. This has large ecological and societal impacts due to crop damage and influences seasonal transmission dynamics of different pathogens [33–35]. The multimammate mouse is a known host for several zoonotic pathogens such as *Lassa mammarenavirus*, *Yersinia pestis*, *Leptospira interrogans*, *Leishmania major* as well as different ecto- and endoparasites [31,36–52]. No studies have investigated or reported on arboviruses in *M. natalensis*, except Diagne et al. (2019) who have detected Usutu virus RNA in *M. natalensis*. However, other studies have reported on sporadic arbovirus detections in other rodent species in sub-Saharan Africa [22,24,53,54]. These findings, along with the ecology of *M. natalensis* (i.e., high abundance during population outbreaks, proximity to humans, and its status as a proven pathogen host) may suggest that this species plays a role in the natural transmission cycle of arboviruses. Consequently, *M. natalensis* could thus pose a risk to humans in east Africa, particularly in Tanzania as an amplifying host.

The human population in Tanzania has experienced several outbreaks of chikungunya virus, Rift Valley fever virus, West Nile virus, and dengue virus in the past decades [55–59]. Due to the symptomatic similarities between arbovirus and malaria infections, which has a prevalence of around 20% in Tanzania, it is probable that arbovirus cases are underreported [60,61]. While these studies confirm that the local human population is indeed exposed to arboviruses, the specific dynamics of arbovirus transmission in this region remains unclear.

The goal of this study was to investigate the potential of wild *M. natalensis* to serve as a host for arboviruses in their natural environment. To achieve this, we first developed a multiplex immune assay to detect immunoglobulin G (IgG) antibodies against 15 different arboviruses and subsequently conducted a comprehensive screening of almost 1,300 blood samples obtained from *M. natalensis* from Morogoro, Tanzania.

## Materials and methods

### Ethics statement

The Ethical Committee for Animal Testing at the University of Antwerp approved the animal experiments performed in this study (ECD2021-79 and ECD2023-08).

## Sample origin

The samples used in this study were collected during previous published and unpublished studies conducted by the University of Antwerp and Pest Management Center of the Sokoine University of Agriculture on *M. natalensis* in Morogoro, Tanzania, between 2010 and 2019 [31,62,63] (Fig 1). The samples were divided in two screening sessions. The first session consisted of approximately 500 dried blood spot (DBS) samples, from wild captured mice that were used in infection and behavioral experiments in six different years (i.e., 2010, 2011, 2015, 2017, 2018, and 2019) with an average of 80 samples per year. The second session consisted of 800 DBS samples from mice involved in capture-mark-recapture experiments in 2017 and 2019. All samples were randomly selected from the studies regardless of individual characteristics or trapping period.

During these studies, *M. natalensis* were live caught using Sherman traps (H.B. Sherman Traps, Tallahassee, USA) in a heterogeneous landscape (e.g., woodlands, maize fields, and fallow land) on the premises of the Sokoine University of Agriculture in Morogoro, Tanzania. Blood was collected from the retro-orbital plexus using a 50μL hematocrit capillary tube and preserved on filter paper (Serobuvard; LDA 22; Zoopole, France). The filter paper was dried for 12 hours at room temperature and archived at -20˚C in envelopes with desiccant. Additional data related to characteristics such as sex, reproductive status, weight, and body measurements were recorded. More detailed Information pertaining to the trapping procedures and sampling methodology can be found in the primary research documents associated with these studies [31,62–64].

## Analysis and protocol

**Assay set up.** To assess the presence of arbovirus antibodies in DBS against a panel of arboviruses, we first developed a multiplex immune assay using Luminex technology [18,65]

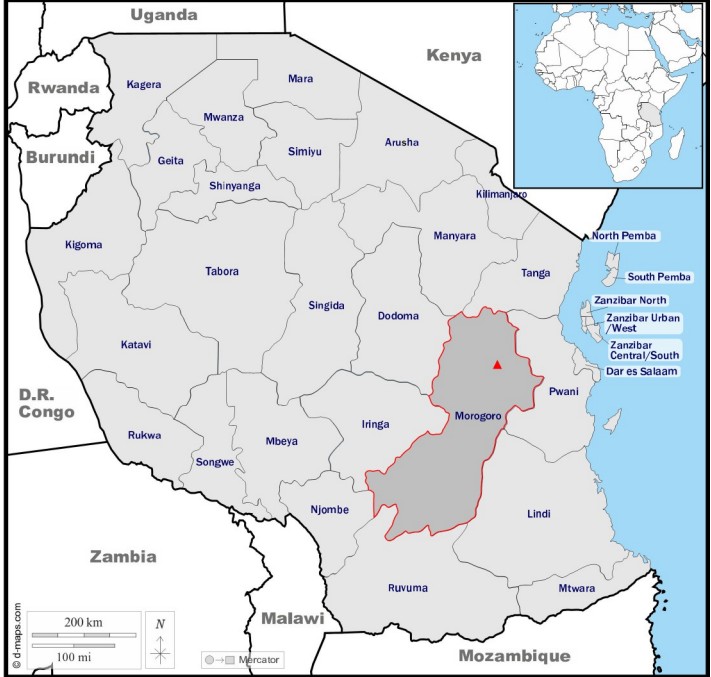

**Fig 1. African continent with a focus on Tanzania.** Samples were collected in the city of Morogoro (red triangle) which is located in the Morogoro region. Basemap origins: https://d-maps.com/carte.php?num_car=736&lang=en and https://d-maps.com/carte.php?num_car=4976&lang=en.

(S1 File). Recombinant virus-derived proteins (Table 1) were covalently coupled to carboxyl-functionalized fluorescent magnetic beads (1–3μg/1.25*10^6 beads) (Luminex Corp. MagPlex-Microspheres; Bio-Rad; Temse, Belgium) employing the BioPlex amine coupling kit (Ref.: 171406001; Bio-Rad; Temse, Belgium) following the manufacturer's instructions.

**Arbovirus protein inoculation.** To obtain positive control samples we inoculated captive *M. natalensis* individuals (age: 5–12 months) from our breeding colony at the University of Antwerp with recombinant virus-derived proteins (Table 1) [66–68]. We subcutaneously injected 4μg of the respective virus protein and 1μL of vaccine adjuvant (Quil-A adjuvant; InvivoGen; Toulouse, France), dissolved in autoclaved phosphate buffered saline (PBS) to achieve a final volume of 1mL. This inoculum was evenly divided, with 0.5mL administered into the scruff and 0.5mL into the hindlimb of the animal, using a 25-gauge, 12.5mm needle and a 0.5mL syringe. This inoculation was duplicated for each viral protein (i.e., performed in two mice) and repeated twice for each mouse (i.e., inoculation on day 0 and day 20). We collected blood, according to the same method as in the previously mentioned studies, every 10 days from day zero until day 30, at day 30 we also collected whole blood from which serum was extracted. Serum from day 30 from individuals were the antibody response increased over time were considered as positive samples. Day 30 had the highest antibody titer in our tests and is also a time point at which IgG antibody development is anticipated to have reached a peak [69,70].

**Table 1. Recombinant arbovirus proteins used for the bead coupling and the inoculation of captive *Mastomys natalensis*.**

| Viral family | Virus | Protein (reference) | Supplier |
|---|---|---|---|
| *Bunyaviridae* | Rift Valley Fever virus (RVFV) | Nucleoprotein (REC31640) | The native antigen company (Kidlington, United Kingdom) |
| *Flaviviridae* | Yellow Fever virus (YFV) | Nonstructural protein 1 (YFV-NS1) | The native antigen company (Kidlington, United Kingdom) |
| *Flaviviridae* | Zika virus (ZIKV) | Nonstructural protein 1 (40544-V07H) | Interchim (Montluçon Cedex, France) |
| *Flaviviridae* | Dengue virus serotype 1 (DENV1) | Nonstructural protein 1 (DEN-004) | Prospecbio (Rehovot, Israel) |
| *Flaviviridae* | Dengue virus serotype 2 (DENV2) | Nonstructural protein 1 (PIP048A) | BioRad (Temse, Belgium) |
| *Flaviviridae* | Dengue virus serotype 3 (DENV3) | Nonstructural protein 1 (DENV3-NS1) | The native antigen company (Kidlington, United Kingdom) |
| *Flaviviridae* | Dengue virus serotype 4 (DENV4) | Nonstructural protein 1 (DENV4-NS1) | The native antigen company (Kidlington, United Kingdom) |
| *Flaviviridae* | Usutu virus (USUV) | Nonstructural protein 1 (Ab218552) | The native antigen company (Kidlington, United Kingdom) |
| *Flaviviridae* | West Nile virus (WNV) | Nonstructural protein 1 (40346-V07H) | Sinobiological (Eschborn, Germany) |
| *Flaviviridae* | Tick-borne Encephalitis virus (TBEV) | Nonstructural protein 1 (TBEV-NS1) | The native antigen company (Kidlington, United Kingdom) |
| *Flaviviridae* | Wesselsbron virus (WSLV) | Nonstructural protein 1 (REC31698) | The native antigen company (Kidlington, United Kingdom) |
| *Nairoviridae* | Crimean Congo Hemorrhagic Fever virus (CCHFV) | Nucleoprotein (REC31639) | The native antigen company (Kidlington, United Kingdom) |
| *Togaviridae* | Chikungunya virus (CHIKV) | Envelope protein 2 (CHI-003) | Prospecbio (Rehovot, Israel) |
| *Togaviridae* | Mayaro virus (MAYV) | Envelope protein 2 (REC31644) | The native antigen company (Kidlington, United Kingdom) |
| *Togaviridae* | O'nyong nyong virus (ONNV) | Envelope protein 2 (B4TG40) | Interchim (Montluçon Cedex, France) |

**Arbovirus IgG antibody screening.**  Screening was done in 96 flat-bottom well plates, each plate contained DBS samples of 80 wild *M. natalensis*, two background controls, eight negative controls, and a six step dilution series (1:200–1:625,000) of a positive pool sample. Each well in the plate contained 50μL of the corresponding sample type. The samples of the wild *M. natalensis* were acquired by placing a punched-out DBS (round, 0.5 cm diameter) in 200 μL of dilution buffer (1% bovine serum albumin, 0.2% Tween-20, 5% fetal calf serum, 45% distilled water, 50% Hypertonic PBS {0.08% $NaH_2PO_4$, 0.25% Na2HPO$_4$, 8.8% NaCl}). One single DBS punch corresponds to approximately 10μL of blood [71]. The punched DBS were left to elute overnight, in a 1.5mL Eppendorf tube, maintained at a temperature of 4˚C on a plate shaker. This elution was considered a 1:100 dilution and was diluted, with dilution buffer, to 1:200 prior to loading in the 96 well plate. This dilution gave the best signal to noise ratio in our preliminary tests and are in line with previous studies [18,72]. The background control was reading buffer (1% bovine serum albumin, 0.05% NaN$_3$, 100% phosphate buffered saline). The eight negative controls were four DBS, treated the same as the wild *M. natalensis* DBS, and four serum samples in a 1:200 dilution. All negative controls originated from the breeding colony at the University of Antwerp. Serum from 15 positive individuals (i.e., one for each inoculated arbovirus antigen) was pooled to create the positive pool sample, each individual serum had a final dilution in the pool of 1:200.

In each well of the 96 well plate, 25μL of bead mixture was added. The bead mixture consisted out of ~1000 protein-coated beads per arbovirus antigen suspended in reading buffer. The bead mixture of the first screening session did not contain ONNV beads.

Plates, containing 50μL of sample and 25μL of bead mixture per well, were incubated for one hour at room temperature, in the dark and on a plate shaker (Heidolph Titrimax 100; VWR; Leuven, Belgium) at 400rpm/min. After incubation, plates underwent washing with dilution buffer using an automated plate washer (Tecan Hydroflex plate washer; Tecan Benelux; Mechelen, Belgium). Subsequently, we added 50μL Biotin anti-mouse IgG (4μg/mL) (Sigma-Aldrich B7022; Merck Life Science; Hoeilaart, Belgium) to each well and incubated for 40 minutes. After another round of washing, we added 50μL of Streptavidin-R-phycoerythrin (1μg/mL) (10655783; Fisher Scientific; Brussel, Belgium) to each well, followed by a 10-minute incubation. The last wash step used reading buffer, and the final bead pellet was resuspended in 150μL of reading buffer. Beads were read on a Bio-Plex 200 System (Bio-Rad; Temse, Belgium). Results were quantified as the median fluorescent intensity (MFI) based on a minimum of 100 beads per antigen, MFI data can be found in S1 Table.

## Data analysis and statistics

All data preparation, analysis and statistical procedures were conducted using R Statistical Software (R version 4.3.3) [73] (S2 File).

**Weight as age classification.**  We used the body weight of the wild-caught *M. natalensis* individuals at the time of sample collection as a rough proxy for age, which we subdivided into three categories based on the 1/3 quantiles of weight; juvenile (5–26.7g), subadult (>26.7–42g) and adult (>42–91g). These weight classes coincide to the expected sexual maturity, with sexual maturity estimated to occur between 30–40g [29,32].

**Inter plate variation.**  To control for variation between different assay plates and testing days, the MFI results were transformed to relative antibody units using the positive dilution series as a standard curve. The MFI result of the positive control starting dilution (i.e., 1:200) was equalized to 3,125 units and each following dilution step was adjusted proportionally (i.e., the final dilution step 1:625,000 corresponded to 1 unit). The results of the two sessions were combined by linear alignment adjustment. This alignment was based on 86 duplicate samples

encompassing the measurable range, allowing the adjustment of the results from the first session.

**Serostatus, cutoff and seroprevalence.** Finally, each sample was categorized as a binary value (i.e., 1 = positive, 0 = negative) for each of the tested arboviruses. This was done based on whether the unit value exceeded the mean cutoff value for that specific arbovirus antigen. Five cutoff values were determined for each arbovirus antigen: I) the mean plus three times the standard deviation of the negative controls (i.e. *'NegCtrl'*) [18,65]; the change-point analysis, using R package *'changepoint'* (version: 2.2.4), calculated at most one changepoint based on the II) mean (i.e. *'CHP.m'*), III) variance (i.e. *'CHP.v'*) and IV) a combination of mean and variance (i.e. *'CHP.mv'*) of wild-caught samples [74,75] and V) the maximum value of an average antibody curve (i.e. *'Recap'*). This curve was based on wild-caught individuals that were recaptured at least three times and showed seroconversion. Seroconversion of an individual was considered when the individual's maximum unit value was at least four-fold the minimum unit value. This four-fold increase is a standard seroconversion confirmation measure in human antibody studies [76]. An average antibody curve, with days as the explanatory variable, was created for each antigen by aligning the maximum unit value of each recaptured seroconverted individual to the same day. The binary results were used to calculate the seroprevalence for each arbovirus along with a 95% confidence interval (CI), using the *'binom.exact'* from the package *'binom'* (version: 1.1.1.1) [77].

**Statistical tests.** The seroprevalence according to the different cutoff methods was compared to the seroprevalence of the antibody curve cutoff using the *'chisq.test'* from the package *'stats'* (version 4.3.3) [73]

As an indication of cross-reactivity in antibody response between the tested arboviruses, pair-wise Pearson correlations were calculated on the binary results, according to the antibody curve cutoff, of all samples using the *'corr.test'* function of R package *'psych'* (version: 2.4.1) [78]. The cross-reactivity in antibody response was visualized using the *'heatmap.2'* function of the R package *'gplots'* (version: 3.1.3) [79].

A generalized linear model (logit link function and binomial error distribution) was constructed with the package *'stat'* (version: 4.3.1), with the response variable being the binary serostatus of each sample [73]. Age (juvenile, subadult and adult), sex and their interaction were included as explanatory variables. The analysis of variance was performed using a likelihood ratio test, with p-values calculated assuming a chi-squared distribution. Pairwise comparison of the seroprevalence was performed between the six combinations of the explanatory variables (two levels of sex and three levels of age), using the *'emmeans'* package (version: 1.8.9) [80]. To prevent reporting statistical findings based on the reliance of an arbitrary p-value of 0.05, we instead present significance in terms of levels of statistical support based on p-values. P-values exceeding 0.1 are labeled as "no" support and values around 0.05 (range $0.1-\geq 0.01$, symbol: *) as "weak" support. "Moderate" support was assigned to p-values clearly below 0.05 (range $< 0.01-\geq 0.001$, symbol: **), while "strong" support is reserved for p-values lower than 0.05 ($< 0.001$, symbol: ***). This representation in terms of statistical support is based on current statistical reporting practices [81].

## Results

In total 1,280 DBS samples were assessed of which 660 were female, consisting of 256 juveniles, 172 subadults and 232 adults, 620 samples were male with 199 juveniles, 313 subadults and 108 adults. Samples of recaptured individuals were considered as individual samples for all analyses.

## Seroprevalence

The seroprevalences according to the different cutoff methods showed at least a weak statistical support for a different seroprevalence compared to the antibody curve seroprevalence for almost all arboviruses. Histograms of the data and seroprevalence for each tested arbovirus antigen according to the different cutoff methods is shown in S1 and S2 Figs. The cutoff value according to the antibody curve based on the recaptured seroconverted individuals was used as the main cutoff value for all further calculations.

The overall arbovirus seroprevalence, defined as at least positive for one of the tested arboviruses, except ONNV, was almost 24% (95% CI: 21.89–26.66%; N = 1280). ONNV was excluded since the samples of the first session were not screened for antibodies against the ONNV antigen. The seroprevalence for *Flaviviridae* was 20% (95% CI: 17.99–22.46%; N = 1280) and for *Togaviridae*, excluding ONNV, almost 7% (95% CI: 5.48–8.32%; N = 1280). Overall, seroprevalences ranged from 0.62% for DENV3 (95% CI: 0.27–1.23%; N = 1280) and MAYV (95% CI: 0.27–1.23%; N = 1280) to 8.44% for DENV2 (95% CI: 6.97–10.10%; N = 1280), see Table 2.

## Pairwise arbovirus serostatus correlation

The correlations in serostatus of samples between the tested arboviruses are visualized in Fig 2. Correlation between two arboviruses is depicted in color scale with the statistical symbol, lower triangle, and the number of positive samples in the upper triangle. The matrix is accompanied by a dendrogram based on the hierarchical clustering of the correlation coefficients.

**Table 2. Total seroprevalence of each arbovirus and virus family in the wild-caught *M. natalensis* sample set.**

| | Seroprevalence (%) | 95% CI (%) | Nr. positive | Cutoff |
|---|---|---|---|---|
| Arbovirus[a] | 24.22 | 21.89–26.66 | 310 | |
| *Bunyaviridae* | | | | |
| RVFV | 2.58 | 1.78–3.60 | 66 | 37.94 |
| *Flaviviridae* | 20.16 | 17.99–22.46 | 258 | |
| YFV | 2.03 | 1.33–2.96 | 26 | 3.45 |
| ZIKV | 3.44 | 2.51–4.59 | 44 | 15.23 |
| DENV1 | 7.58 | 6.19–9.17 | 97 | 42.11 |
| DENV2 | 8.44 | 6.97–10.10 | 108 | 101.09 |
| DENV3 | 0.62 | 0.27–1.23 | 8 | 32.27 |
| DENV4 | 5.62 | 4.43–7.03 | 72 | 40.66 |
| USUV | 4.61 | 3.53–5.91 | 59 | 6.47 |
| WNV | 2.42 | 1.65–3.42 | 31 | 25.60 |
| TBEV | 1.64 | 1.02–2.50 | 21 | 50.85 |
| WSLV | 3.52 | 2.58–4.68 | 45 | 4.16 |
| *Nairoviridae* | | | | |
| CCHFV | 2.81 | 1.98–3.87 | 36 | 5513.61 |
| *Togaviridae*[a] | 6.80 | 5.48–8.32 | 87 | |
| CHIKV | 6.17 | 4.92–7.63 | 79 | 43.95 |
| MAYV | 0.62 | 0.27–1.23 | 8 | 11.91 |
| ONNV | 2.18 | 1.30–3.42 | 18 | 77.10 |

A 95% confidence interval (CI) is provided, and the calculated cutoff value is in units. Sample size was 1280 for each tested arbovirus except for ONNV which had 826 samples

[a] Indicates that ONNV was not included for that calculation.

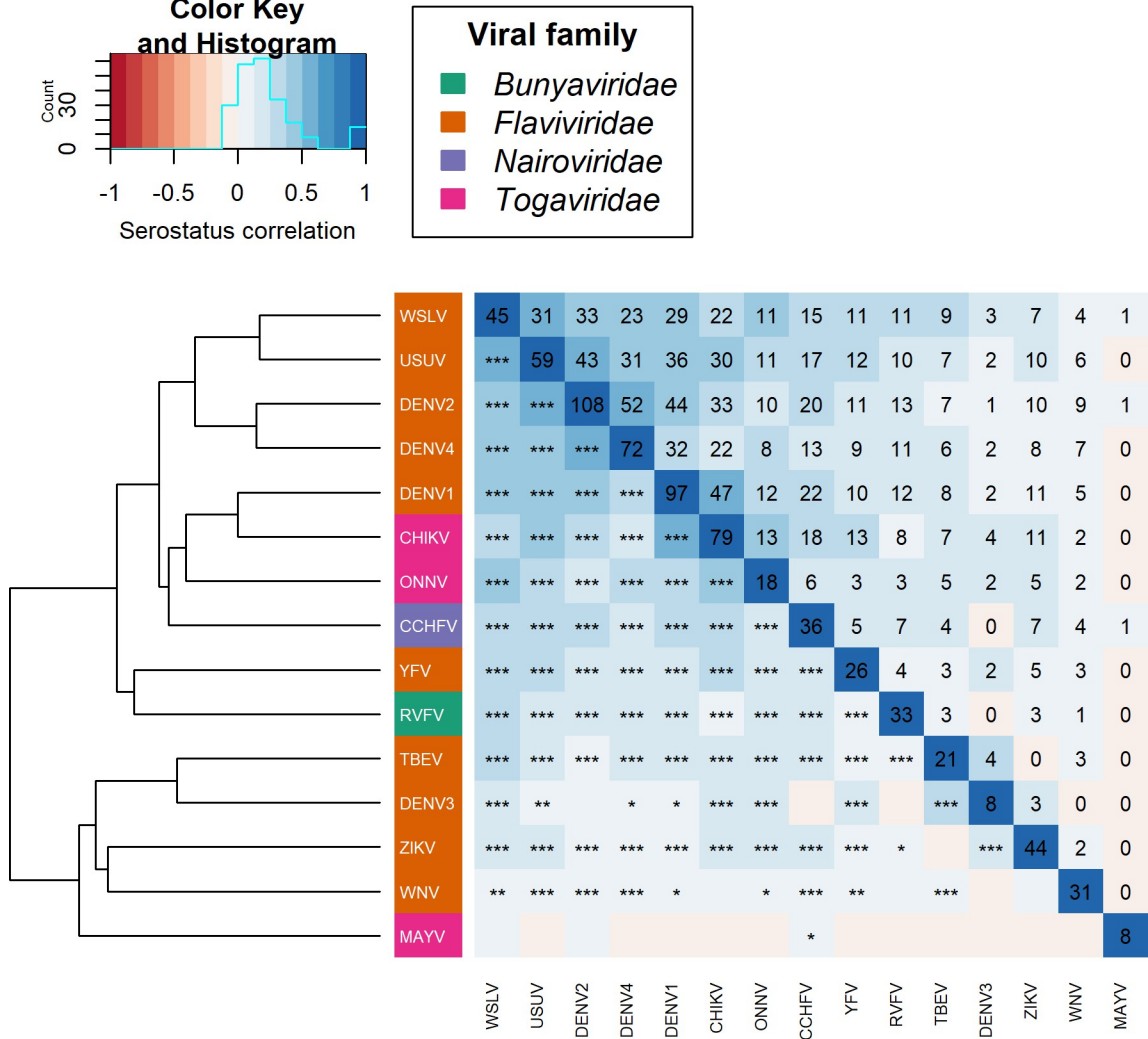

**Fig 2. Correlation of the serostatus response between the tested arboviruses with a dendrogram of hierarchical clustering.**
Symbols in lower triangle represent significance of correlation, values in upper triangle, including diagonal line, represents the number of positive individuals for the corresponding arboviruses.

The branch lengths are a proxy for relative distance between arboviruses based on the serostatus response of the samples.

The correlation in sample response between the tested arboviruses ranged from -2.44% for TBEV and ZIKV with no statistical support (p = 0.384) to almost 59% between WSLV and USUV with a strong statistical support (p< 0.001). The dendrogram based on the hierarchical clustering of the correlation showed that WSLV–USUV (correlation = 58.52%; p< 0.001), DENV2 –DENV4 (correlation = 56.03%; p< 0.001) and DENV1 –CHIKV (correlation = 50.31%; p< 0.001) are relatively closer to each other than to other tested arboviruses.

## Model analysis of antibody response

**Sex, age, and interaction effects on serostatus.** The generalized linear model indicated that there was a weak statistical interaction between the effects of sex and age on serostatus for DENV4 (Deviance [Df. = 2; Res.Df. = 1274] = 5.44; p = 0.066) and USUV (Deviance [Df. = 2;

Res.Df. = 1274] = 7.72; p = 0.021). For the other tested arboviruses, no support for a statistical interaction was detected, the interaction was thus removed from those models. In the case of RVFV, no statistical support was found for an effect of sex, age, or the interaction on the serostatus. All results from generalized linear model's analysis of variance are reported in S2 Table.

The analysis of the sex variable showed a moderate support for males having a lower seroprevalence compared to females for DENV2 (Est. $_{males}$ ± SE = -0.665 ± 0.242; p = 0.006) and WSLV (Est. $_{males}$ ± SE = -1.446 ± 0.483; p = 0.003). A weak statistical effect of a lower seroprevalence in males compared to females was detected in ZIKV (Est. $_{males}$ ± SE = -0.643 ± 0.386; p = 0.096), TBEV (Est. $_{males}$ ± SE = -1.103 ± 0.635; p = 0.083), CCHFV (Est. $_{males}$ ± SE = -0.821 ± 0.460; p = 0.074) and CHIKV (Est. $_{males}$ ± SE = -0.633 ± 0.296; p = 0.032). There was no support for a difference in seroprevalence between males and females for RVFV (Est. $_{males}$ ± SE = 0.097 ± 0.381; p = 0.798), YFV (Est. $_{males}$ ± SE = 0.425 ± 0.413; p = 0.304), DENV1 (Est. $_{males}$ ± SE = -0.377 ± 0.248; p = 0.129), DENV3 (Est. $_{males}$ ± SE = -17.43 ± 2021.76; p = 0.993), WNV (Est. $_{males}$ ± SE = -0.009 ± 0.397; p = 0.982), MAYV (Est. $_{males}$ ± SE = -1.222 ± 1.080; p = 0.258) and ONNV (Est. $_{males}$ ± SE = -17.75 ± 1663.70; p = 0.991).

The analysis of the age variable showed a strong statistical support for a higher seroprevalence in subadults than in juveniles for DENV1 and DENV2, a moderate support for CHIKV and a weak support for YFV, ZIKV, TBEV, WSLV and CCHFV. There was no support for a difference in subadult and juvenile seroprevalence in the other tested arboviruses. A significantly higher seroprevalence in adults compared to juveniles was shown for ZIKV, DENV1, DENV2, WSLV and CHIKV with a strong support. A moderate support for a higher seroprevalence in adults than in juveniles was detected for YFV and CCHFV. Adults showed a weak statistical support for a higher seroprevalence in contrast to juveniles for WNV and TBEV. All other tested arboviruses showed no support for a statistical difference between adults and juveniles. The comparison between subadults and adults showed a strongly supported statistical difference for DENV1 and CHIKV with a higher seroprevalence in adults. A moderate support for a higher seroprevalence in adults compared to subadults was detected for ZIKV and CCHFV. Yellow Fever virus, WNV, WSLV and ONNV showed a weak support for a statistically higher seroprevalence in adults than in subadults. The other arboviruses showed no statistically significant difference between adults and subadults. See Table 3 for estimates, standard errors, and p-values.

In the case of DENV4, there was a weak support for an interaction between the effects of age and sex: the analysis showed that there was a strong statistical support for a higher seroprevalence in female adults compared to female juveniles (Est. $_{female—adult}$ ± SE = 1.844 ± 0.486; p< 0.001). A moderate support was shown for a higher seroprevalence in female subadults compared to female juveniles (Est. $_{female—subadult}$ ± SE = 1.379 ± 0.517; p = 0.008) and a weak support for a higher seroprevalence in female adults compared to male adults (Est. $_{female—adult}$ ± SE = 1.898 ± 0.736; p = 0.010). For USUV the model analysis with a weak interaction, showed that there was weak statistical support for a higher seroprevalence in female adults compared to female subadults (Est. $_{female—adult}$ ± SE = 0.868 ± 0.345; p = 0.012) and also a weak support for a higher seroprevalence in female adults compared to male adults (Est. $_{female—adult}$ ± SE = 1.515 ± 0.612; p = 0.013).

Fig 3 displays the seroprevalence for the six distinct levels (two levels of sex and three levels of age) for all tested arboviruses, with statistical support lines based on the log odds.

## Discussion

In this study, we optimized a high-throughput multiplex immunoassay for the simultaneous detection of IgG antibodies against 15 medically relevant arboviruses and used it to investigate

**Table 3. Difference in coefficient estimate on logit scale between the age levels with standard error (SE).**

| | Juvenile—Subadult | | Juvenile—Adult | | Subadult—Adult | |
|---|---|---|---|---|---|---|
| | Estimate ± SE | p-value | Estimate ± SE | p-value | Estimate ± SE | p-value |
| RVFV | -0.227 ± 0.480 | 0.637 | -0.649 ± 0.446 | 0.146 | -0.422 ± 0.421 | 0.316 |
| YFV | -1.368 ± 0.794 | **0.085** | -2.197 ± 0.757 | **0.004** | -0.829 ± 0.449 | **0.065** |
| ZIKV | -1.349 ± 0.656 | **0.040** | -2.308 ± 0.611 | **< 0.001** | -0.959 ± 0.365 | **0.009** |
| DENV1 | -1.610 ± 0.456 | **< 0.001** | -2.468 ± 0.434 | **< 0.001** | -0.858 ± 0.243 | **< 0.001** |
| DENV2 | -1.551 ± 0.356 | **< 0.001** | -1.780 ± 0.352 | **< 0.001** | -0.229 ± 0.221 | 0.301 |
| DENV3 | -16.47 ± 2176.65 | 0.994 | -18.14 ± 2176.65 | 0.993 | -1.665 ± 1.073 | 0.121 |
| WNV | 0.007 ± 0.582 | 0.991 | -1.20 ± 0.477 | **0.012** | -1.207 ± 0.480 | **0.012** |
| TBEV | -1.855 ± 1.083 | **0.087** | -2.566 ± 1.039 | **0.014** | -0.711 ± 0.499 | 0.155 |
| WSLV | -0.960 ± 0.539 | **0.075** | -1.656 ± 0.492 | **< 0.001** | -0.697 ± 0.357 | **0.051** |
| CCHFV | -1.999 ± 1.072 | **0.062** | -3.321 ± 1.021 | **0.001** | -1.321 ± 0.434 | **0.002** |
| CHIKV | -2.267 ± 0.749 | **0.002** | -3.474 ± 0.723 | **< 0.001** | -1.207 ± 0.283 | **< 0.001** |
| MAYV | 0.642 ± 1.228 | 0.601 | -0.807 ± 0.844 | 0.339 | -1.449 ± 1.104 | 0.189 |
| ONNV | -17.05 ± 1747.82 | 0.992 | -18.15 ± 1747.82 | 0.992 | -1.099 ± 0.641 | **0.087** |

Data originates from the pairwise comparison of the age class variables of the generalized linear model. P-values marked in bold have at least a weak statistical support (p< 0.1).

the potential of *M. natalensis* to serve as a host for arboviruses. Besides the high throughput and multiplexing possibility, bead-based Luminex assays come with several other advantages, such as the small sample volumes required for testing, the compatibility with diverse sample types, incl. eluted DBS, the cost-effectiveness with lower reagent cost and reduced labor time, a broad dynamic range for accurate quantification, a reduced variability as a result of running multiple analytes in a single assay, and the versatility and flexibility of customizable panels. We describe the screening results of an archived set of wild *M. natalensis* DBS samples. Our results revealed an overall seroprevalence of 24% against the entire panel of tested antigens. Virus family-specific seroprevalences were approximately 2.6%, 20%, 2.8% and 7% for respectively *Bunyaviridae*, *Flaviviridae*, *Nairoviridae* and *Togaviridae*. We further found that female rodents were more likely to be classified as antibody positive for eight of the 15 tested arboviruses. Additionally, positivity increased significantly with age for almost all tested arboviruses.

The lack of realistic natural positive controls limits the possibility to calculate the assay's sensitivity and determine a true cutoff, we therefore used recaptured seroconverted individuals to determine a cutoff value. The use of antibody titers at multiple time points are a standard practice to determine antibody or pathogen development and (sero)conversions in human studies [76]. However, multiple samples of an individual animal across time are often impossible or very difficult in wildlife studies. Our study is unique in that regard that we have measurements of individual recaptured *M. natalensis*. We consider that our cutoff based on seroconverted individuals is a good proxy for the natural cutoff value, since it is based on similar methods as in human studies [76]. We tried to show in our analysis that the tested mathematical methods could approximate this calculated cutoff and thus provide a method for future studies that do not have access to recaptured seroconverted wildlife samples. Unfortunately, the tested cutoff methods did not significantly approximate the seroprevalence according to the cutoff using samples from recaptures. The negative control-based cutoff (i.e., the mean plus three times the standard deviation of the negative control samples) gave unrealistic high seroprevalences. This can be explained by the fact that the negative control samples originate from a breeding colony and could thus also not be used to determine the assay's

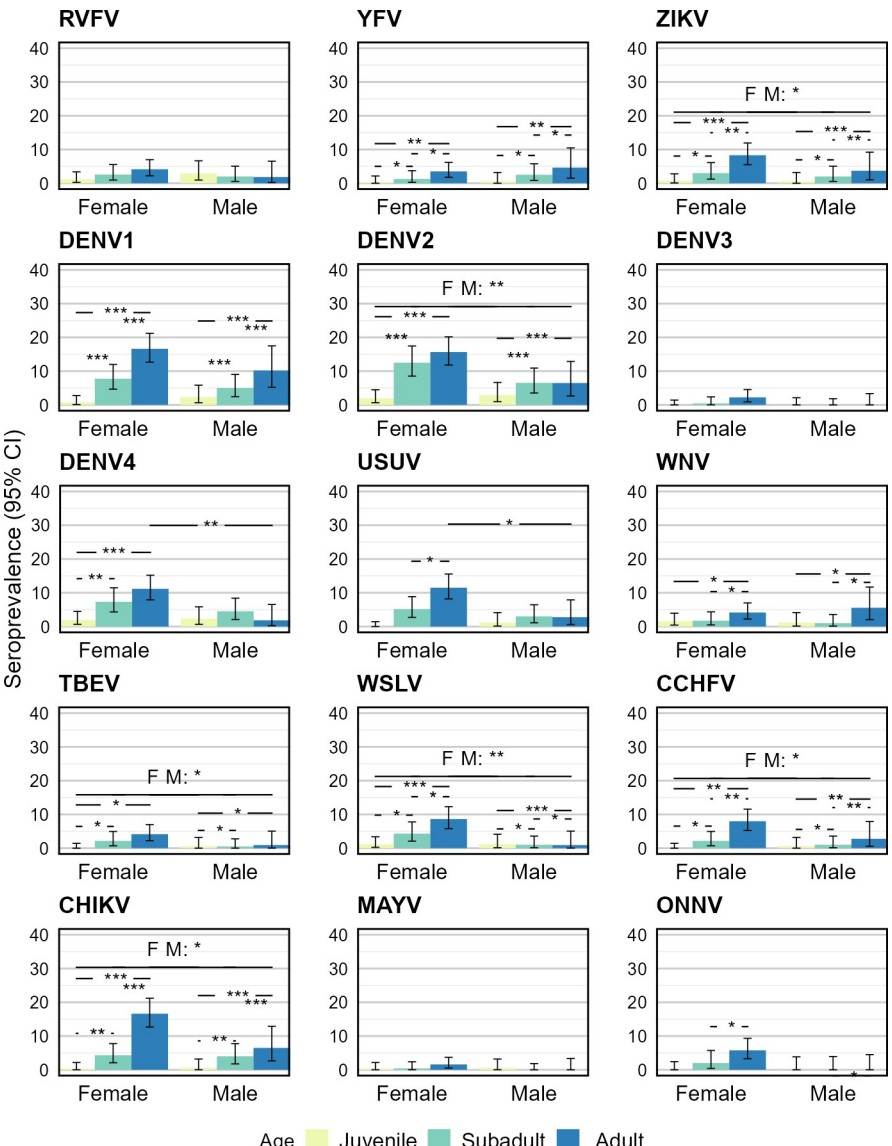

**Fig 3. Seroprevalence according to sex and age combinations with 95% confidence error bars for each arbovirus.**
Statistical support on seroprevalence difference is indicated by asterisks in the horizontal lines. Sample size: 660
females of which 256 juveniles, 172 subadults and 232 adult and 620 males with 199 juveniles, 313 subadults and 108
adults. Sample sizes for ONNV are 560 females: 150 juveniles, 150 subadults, 260 adults and 266 males: 94 juveniles, 92
subadults and 80 adults.

specificity. These animals have thus never been exposed to a natural environment and the
pathogens that occur in the environment.

The statistical methods vary in their seroprevalence with some methods approximating the
estimated seroprevalence according to the recaptured cutoff. This high degree of variation
makes it difficult to decide on one method that works for all the tested arboviruses. The cutoff
value for CCHFV seems extremely high compared to the other arboviruses, but the unit values
for CCHFV are also much higher than for the other arboviruses (see S1 Fig). The reason is that
the unit values are calculated based on the positive dilution series and the positive controls for
CCHFV were not of the same magnitude as for the other arboviruses. The value in

determining a cutoff and the resulting seroprevalence is that it allows the comparison of results with previous and future studies on arbovirus seroprevalence in rodents or other wildlife. We are aware that the used cutoff and resulting seroprevalences could be an over- or underestimation and might not reflect the natural arbovirus seroprevalence. We therefore encourage future research to investigate and compare different cutoff methods for arbovirus (or pathogen) antibody detection in wildlife studies.

The detection of antibodies against each of 15 tested arbovirus antigens indicates that these arboviruses, or closely related viruses, are present in *M. natalensis*. The overall arbovirus seroprevalence of 24% suggests that this rodent species is commonly infected with one or more arboviruses and that it could thus play a significant role in virus transmission and persistence. Our results corroborate previous studies, which detected USUV and WNV RNA in respectively *M. natalensis* and *M. erythroleucus*, in Senegal [24,54]. Besides in this genus, arboviral RNA has also been found in other rodents in Africa, such as *Rattus rattus* for USUV and WSLV and *Desmodillus auricularis* for WSLV [22,24,53]. The findings in our study thus further corroborate that arboviruses are likely present in rodents, and specifically in the ubiquitous *M. natalensis*. The demographic and ecological characteristics of *M. natalensis* may have particularly important implications for arbovirus transmission. The population densities of *M. natalensis* in Tanzania are strongly dependent on weather conditions. More specifically, early rainfall and elevated temperatures lead to an exponential growth in the population density, due to an increase influx of juveniles [29,30,32]. The rainfall and increased temperatures are also beneficial for the breeding of mosquitoes and the multiplication of arboviruses within these vectors [82]. Further, *M. natalensis* is highly abundant around houses and in the crop fields at the fringes of the villages. These factors increase the likelihood of arbovirus outbreaks in *M. natalensis* populations, with the possibility of spillover to humans.

Arboviruses that show the highest seroprevalence are DENV1, DENV2, DENV4 and CHIKV, with seroprevalences between five to nine percent. These seroprevalences could be caused by cross-reactivity due to antibodies of other dengue virus serotypes or other flaviviruses binding to the non-structural protein 1 (NS1 protein) of DENV1, DENV2 and DENV4. The same effect could also be true for alphaviruses binding to the envelope protein 2 (E2 protein) of CHIKV. Whether these seroprevalences are indeed due to the presence of the arbovirus specific antibodies or a related arbovirus remains to be investigated. Nonetheless, it indicates that a part of the sampled *M. natalensis* population in Morogoro is exposed to dengue virus and CHIKV or respectively to a related flavivirus and alphavirus. This hypothesis is supported by the fact that flavi- and alphaviruses are the most prevalent arboviral genera in humans, compared to other arbovirus genera, and potentially thus also in rodents involved in the sylvatic cycle [83,84].

A recent health survey has shown that, in our samples' region of origin, a high percentage of the human population is seropositive for CHIKV (9.83%) [55]. Another study in the same region reported acute infection of CHIKV in 1.28% of patients with fever and malaria-like symptoms [85]. Although these studies have not found any indication of dengue virus in humans, a large-scale cross-sectional study in Tanzania has found CHIKV and dengue virus antibodies in respectively 28.0% and 16.1% of the population [56]. These studies clearly indicate that the human population in Tanzania is exposed to arboviruses and then specifically to CHIKV and DENV.

The cross-reactivity analysis via the correlation matrix and hierarchical clustering (Fig 2) showed an antibody response correlation between WSLV–USUV (59%), DENV2 –DENV4 (56%) and DENV1 –CHIKV (50%). We expected that phylogenetically related arboviruses would show elevated levels of correlation due to cross-reactivity [86]. A remarkable result in this cross-reactivity analysis is that DENV1 –CHIKV cluster together with a correlation of

50%, based on the serostatus of the tested samples. The branch DENV1/CHIKV clusters also closer to ONNV than to the branches of WSLV/USUV and DENV2/DENV4. This is unexpected since CHIKV belongs to the *Togaviridae* and DENV1 to the *Flaviviridae* [87]. The proteins used for the antibody detection are also two different proteins, with the E2 protein used for the *Togaviridae* and NS1 protein for the *Flaviviridae*, thus limiting the possibility of cross-reactivity. Although we cannot exclude that there might be similar epitopes between the different proteins, other studies have already indicated that cross-reactivity between the E2 protein of the *Togaviridae* and NS1 protein of the *Flaviviridae* is limited [88,89]. The residue identity between the two proteins is also less then 13% according to the amino acid alignment algorithm of Geneious Prime. Given that both *Togaviridae* and *Flaviviridae* viruses are circulating in humans in East Africa, we hypothesize that these viral families may also both be present in rodents [5,10]. More specifically, it is plausible that both viral families could be found in *M. natalensis*, where pathogen co-infections are common [49]. This hypothesis is further supported by the fact that some viruses in both families are transmitted by the same arthropod vectors, such as *Aedes aegypti* and *Aedes albopictus* for both dengue virus and CHIKV [4,82].

For some of the tested arboviruses, we found statistical support for a higher seroprevalence in females than in males. This result is supported by previous studies where it is shown that female mice have a stronger innate immune response than male mice [90]. In other animals (e.g., birds, fish, insects) as well as humans, females also display stronger immune responses [91–95]. The major driving forces behind these immune differences are genetic (i.e., X-chromosome-linked) and hormonal (i.e., different estrogen and testosterone levels) [96]. In the case of *M. natalensis*, behavioral differences could also be the cause for this divergence in seroprevalence. Previous studies have already shown that home range, behavior and pathogen presence differ between male and female *M. natalensis* [48,97]. These inherent sex differences in sensitivity to infections could influence the seroprevalence, where the calculated cutoff value could be an over or under estimation for a particular sex. However, since our sample size of males (N = 620) and females (N = 660) is approximately the same this influence was considered to be insignificant. Besides the sex effects, we also found statistical support for a positive age effect on the presence of antibodies in some of the tested arboviruses. This increased seroprevalence with age corroborates previous findings for other pathogens (i.e., *Bartonella sp.*, *Anaplasma sp.*, helminths, and arenaviruses) [48,49,98]. This age effect further supports our hypothesis that *M. natalensis* is exposed to arboviruses and that individuals develop antibodies and gain immunity via repeated exposures throughout their life. To maintain the arbovirus transmission in the *M. natalensis* population, there needs to be a proportion of the population that is either chronically infected or immunologically naïve. Chronic infections in *M. natalensis* have already been documented for mammarenaviruses [31,99]. However, as far as we are aware, naturally occurring chronic arbovirus infections have not been reported in humans or non-human vertebrates. Therefore, the presumable driving factor in sustained transmission is the presence of immunologically naïve individuals. During the breeding season, which coincides with increased rainfall and temperature, there is an influx of immunologically naïve juveniles. This influx can reach high proportions during population outbreak periods [30,64]. We thus expect that it is juveniles who are the major factor in sustaining the arbovirus transmission cycle. We predict that the prevalence of arboviral genetic material will be higher in juveniles than in adults, since juveniles do not possess the necessary antibodies to fight of the infection.

We conclude from our detected antibody responses that arboviruses, or related viruses, are present in *M. natalensis* in Morogoro, Tanzania. The higher seroprevalence we detect in females can be explained by genetic, hormonal, ecological and/or behavioral differences between sexes. Individuals are exposed to these viruses throughout their life and gain

immunity as they age. We hypothesize that juvenile *M. natalensis* play an essential role in sustaining arbovirus transmission as they are immunologically naïve and can reach high densities in favorable climate conditions that coincide with optimal vector conditions. More extensive screening, such as virus neutralization tests and molecular screening of these viruses within *M. natalensis* are necessary to quantify the contribution of this rodent species in the arbovirus transmission cycle.

## Supporting information

**S1 File. Standard operating procedure (SOP) of the developed high-throughput multiplex immunoassay to detect antibodies against 15 medically relevant arboviruses using Luminex technology.**
(PDF)

**S2 File. RMarkdown file detailing the statistical analysis used and described in this manuscript.**
(HTML)

**S1 Fig. Histograms of wild-caught *M. natalensis* for each tested arbovirus with on the x-axis the relative antibody units in a logarithmic scale.** The relative antibody units are calculated according to the positive control dilution series. The calculated cutoff values are represented by the colored vertical lines: 'CHP.m' is the changepoint mean, 'CHP.mv' is the changepoint mean-variance, 'CHP.v' is the changepoint variance, 'NegCtrl' is the mean plus three times the standard deviation of the negative control samples and 'Recap' is the maximum value of an antibody development curve based on recaptured seroconverted wild-caught *M. natalensis.*
(SVG)

**S2 Fig. Estimated seroprevalence, according to the calculated cutoff methods, of the wild-caught *M. natalensis* with 95% confidence interval for each of the tested arboviruses.** The cutoff methods: 'CHP.m' is the changepoint mean, 'CHP.mv' is the changepoint mean-variance, 'CHP.v' is the changepoint variance, 'NegCtrl' is the mean plus three times the standard deviation of the negative control samples and 'Recap' is the maximum value of an antibody development curve based on recaptured seroconverted wild-caught *M. natalensis.* Each calculated seroprevalence was compared to the 'Recap' seroprevalence using a Chi-square test, significant difference is depicted in asterisk (*) symbols. P-values: * $0.1-\geq 0.01$; ** $< 0.01-\geq 0.001$; *** $< 0.001$.
(SVG)

**S1 Table. Raw MFI data of all the samples used and described in this manuscript.**
(XLSX)

**S2 Table. Analysis of variance from the generalized linear model (logit link function and binomial error distribution) with the response variable being the binary serostatus of each sample.** Sex and age and their interaction were included as explanatory variables. P values with at least a weak statistical support are marked in bold ($p< 0.1$).
(XLSX)

## Author Contributions

**Conceptualization:** Wim De Kesel, Martine Peeters, Erik Verheyen, Joachim Mariën, Kevin K. Ariën.

**Data curation:** Wim De Kesel, Bram Vanden Broecke, Benny Borremans, Joachim Mariën, Kevin K. Ariën.

**Formal analysis:** Wim De Kesel.

**Funding acquisition:** Wim De Kesel, Martine Peeters, Erik Verheyen, Kevin K. Ariën.

**Investigation:** Wim De Kesel, Léa Fourchault, Elisabeth Willems, Ann Ceulemans.

**Project administration:** Wim De Kesel, Herwig Leirs, Sophie Gryseels, Joachim Mariën, Kevin K. Ariën.

**Resources:** Wim De Kesel, Bram Vanden Broecke, Benny Borremans, Christopher Sabuni, Apia Massawe, Rhodes H. Makundi, Herwig Leirs, Joachim Mariën, Kevin K. Ariën.

**Software:** Wim De Kesel.

**Supervision:** Erik Verheyen, Sophie Gryseels, Joachim Mariën, Kevin K. Ariën.

**Visualization:** Wim De Kesel.

**Writing – original draft:** Wim De Kesel.

**Writing – review & editing:** Wim De Kesel, Bram Vanden Broecke, Benny Borremans, Léa Fourchault, Martine Peeters, Erik Verheyen, Sophie Gryseels, Joachim Mariën, Kevin K. Ariën.

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
