## [Decision Letter · Decision Letter 0]

16 Jul 2024

Dear Mr. De Kesel,

Thank you very much for submitting your manuscript "Antibodies against medically relevant arthropod-borne viruses in the ubiquitous African rodent Mastomys natalensis" for consideration at PLOS Neglected Tropical Diseases. As with all papers reviewed by the journal, your manuscript was reviewed by members of the editorial board and by several independent reviewers. The reviewers appreciated the attention to an important topic. Based on the reviews, we are likely to accept this manuscript for publication, providing that you modify the manuscript according to the review recommendations. 

Sincerely,

Ran Wang, M.D.

Academic Editor

Michael Holbrook

Section Editor

Reviewer's Responses to Questions

**Key Review Criteria Required for Acceptance?**

**Methods**

-Are the objectives of the study clearly articulated with a clear testable hypothesis stated?

-Is the study design appropriate to address the stated objectives?

-Is the population clearly described and appropriate for the hypothesis being tested?

-Is the sample size sufficient to ensure adequate power to address the hypothesis being tested?

-Were correct statistical analysis used to support conclusions?

-Are there concerns about ethical or regulatory requirements being met?

Reviewer #1: -Are the objectives of the study clearly articulated with a clear testable hypothesis stated? Yes

-Is the study design appropriate to address the stated objectives? Yes

-Is the population clearly described and appropriate for the hypothesis being tested? Yes

-Is the sample size sufficient to ensure adequate power to address the hypothesis being tested? Yes

-Were correct statistical analysis used to support conclusions? Yes

-Are there concerns about ethical or regulatory requirements being met? No

Reviewer #2: (No Response)

**Results**

-Does the analysis presented match the analysis plan?

-Are the results clearly and completely presented?

-Are the figures (Tables, Images) of sufficient quality for clarity?

Reviewer #1: The analysis presented match the analysis plan and results are clearly presented using good quality figures.

Reviewer #2: 1. Page 18, line 348 - Please correct grammar.

2. Page 23, line 453-458 - The authors showed that the DBS samples were cross-reactive with different viral proteins like NS1 of DENV and E2 of CHIKV, and they discussed the cross-reactivity to between different two proteins. So to well understand that point, please indicate the amino acid homology rate between NS1 and E2 proteins.

3. Common experimental mouse has different sensitivity to viral infections between sex. Do authors think the sensitivity affect the seropositivity in your results?

4. Authors showed the seroprevalence of Arbovirus you listed. Are there reports that those viruses experimentally infect to mouse, and induced antibodies to NS1 or E2 proteins?

5. Please correct Fig 2, especially Zika virus. Authors Zika virus belongs to Togaviridae. This is not correct.x

6. How is the sensitivity of this high-throughput multiplex immune assay?

**Conclusions**

-Are the conclusions supported by the data presented?

-Are the limitations of analysis clearly described?

-Do the authors discuss how these data can be helpful to advance our understanding of the topic under study?

-Is public health relevance addressed?

Reviewer #1: The authors conclude that M.natalensis play an important role in sustaining arbovirus transmission and this is supported by the data presented in the results (an overall seroprevalence of 24%). The paper clearly stated the limitations in their analysis such as using negative control from a breeding colony that could potentially over/under-estimate the sero-prevalence. 

Do the authors discuss how these data can be helpful to advance our understanding of the topic under study? Yes

Is public health relevance addressed? Yes

Reviewer #2: (No Response)

**Editorial and Data Presentation Modifications?**

Reviewer #1: Though the paper is well-written i think some background on the multiplex immunoassay will help to understand why the authors chose this method.

The method section is well-written however it would read better if the authors could consider adding subtitles to subsections e.g. Screening for IgG antibodies to arboviruses, recombinant arbovirus proteins etc

More details on beads coupling (if it was done)

Reviewer #2: (No Response)

**Summary and General Comments**

Reviewer #1: De Kesel and co-authors present an interesting study that address a key gap in the existing knowledge of arboviral diseases transmission dynamics. The authors hypothesize that rodents specifically M.natalensis may serve as amplifying hosts of arboviruses. To this end, they screened blood samples from M.natalensis individuals for antibodies against 15 medically important arboviruses and reported an overall arbovirus seroprevalence of 24% suggesting that M.natalensis plays a role in the transmission of multiple arboviruses. The main strength of this paper is that the authors chose a perfect rodent species to study arboviruses transmission dynamics. Besides being a reservoir host of numerous pathogens and also the most abundant rodent species in SSA, M.natalensis lives in close proximity to humans increasing the risk of pathogen spillover to humans. As such, the paper identifies rodents as potential sources of arbovirus infections that pose a threat to human health. One of the weaknesses or rather a limitation is that the authors used negative controls obtained from a breeding colony instead of natural positive controls which could have affected the cut-off values. I enjoyed reading this manuscript and i think the paper is well written. However, some background information on the immunoassay used is missing and the analysis, set up and protocol in the methods section could be strctured into subsections with subtitles.

Reviewer #2: The authors developed a high-throughput multiplex immune assay using Luminex technology. The system employed the viral proteins such as NS1 protein for Flaviviridae and E2 protein for Togaviridae to evaluate the seroprevalence for 15 species of arboviruses in multimammate mouse and Mastomysnatalensis in Tanzania. They showed the these african small mammals had highly seropositive rate to several arboviruses, such as Dengue virus type 2 (8.44%), Usutu virus (4.61%), Chikungunya virus (6.17%), and etc. Moreover, they presented the difference in seropositivity between sex or age. These results suggest that these small mammals play a role of transmission of arboviruses in African continent. This work is necessary to understand the viral circulations. This accumulates our knowledge to maintain public health.

If the authors continue this study, I'd like to try to isolate the arboviruses from African small animals.

PLOS authors have the option to publish the peer review history of their article (what does this mean?). If published, this will include your full peer review and any attached files.

Reviewer #1: No

Reviewer #2: No

Figure Files:

Data Requirements:

Reproducibility:

References

---

## [Decision Letter · Decision Letter 1]

20 Aug 2024

Dear Mr. De Kesel,

We are pleased to inform you that your manuscript 'Antibodies against medically relevant arthropod-borne viruses in the ubiquitous African rodent Mastomys natalensis' has been provisionally accepted for publication in PLOS Neglected Tropical Diseases.

Best regards,

Ran Wang, M.D.

Academic Editor

Michael Holbrook

Section Editor

Reviewer's Responses to Questions

**Key Review Criteria Required for Acceptance?**

**Methods**

-Are the objectives of the study clearly articulated with a clear testable hypothesis stated?

-Is the study design appropriate to address the stated objectives?

-Is the population clearly described and appropriate for the hypothesis being tested?

-Is the sample size sufficient to ensure adequate power to address the hypothesis being tested?

-Were correct statistical analysis used to support conclusions?

-Are there concerns about ethical or regulatory requirements being met?

Reviewer #1: yes

Reviewer #2: -Are the objectives of the study clearly articulated with a clear testable hypothesis stated? Yes.

-Is the study design appropriate to address the stated objectives? Yes.

-Is the population clearly described and appropriate for the hypothesis being tested? Yes.

-Is the sample size sufficient to ensure adequate power to address the hypothesis being tested? Yes.

-Were correct statistical analysis used to support conclusions? Yes.

-Are there concerns about ethical or regulatory requirements being met? Yes.

**Results**

-Does the analysis presented match the analysis plan?

-Are the results clearly and completely presented?

-Are the figures (Tables, Images) of sufficient quality for clarity?

Reviewer #1: yes

Reviewer #2: -Does the analysis presented match the analysis plan? Yes.

-Are the results clearly and completely presented? Yes.

-Are the figures (Tables, Images) of sufficient quality for clarity? Yes.

**Conclusions**

-Are the conclusions supported by the data presented?

-Are the limitations of analysis clearly described?

-Do the authors discuss how these data can be helpful to advance our understanding of the topic under study?

-Is public health relevance addressed?

Reviewer #1: yes

Reviewer #2: -Are the conclusions supported by the data presented? Yes.

-Are the limitations of analysis clearly described? Yes.

-Do the authors discuss how these data can be helpful to advance our understanding of the topic under study? Yes.

-Is public health relevance addressed? Yes.

**Editorial and Data Presentation Modifications?**

Reviewer #1: Accept

Reviewer #2: The authors have responded appropriately to the reviewers' comments, and their replies have enabled us to fully understand this paper. I recommend that this paper be accepted.

**Summary and General Comments**

Reviewer #1: All comments are addressed in a satisfactory manner.

Reviewer #2: I don't have no more comments. This paper is well written.

PLOS authors have the option to publish the peer review history of their article (what does this mean?). If published, this will include your full peer review and any attached files.

Reviewer #1: No

Reviewer #2: No

---

## [Editor Report · Acceptance letter]

29 Aug 2024

Dear Mr. De Kesel,

We are delighted to inform you that your manuscript, "Antibodies against medically relevant arthropod-borne viruses in the ubiquitous African rodent Mastomys natalensis," has been formally accepted for publication in PLOS Neglected Tropical Diseases.

Best regards,

Shaden Kamhawi

co-Editor-in-Chief

Paul Brindley

co-Editor-in-Chief
